# Periodicity in Steller's eider (*Polysticta stelleri*) population size and density on the Arctic Coastal Plain, Alaska, revealed using generalized additive models

Erik E. Osnas◉*

Division of Migratory Bird Management, United States of America Fish and Wildlife Service, Anchorage, Alaska, United States of America

* erik_osnas@fws.gov

## Abstract

Annual population and spatial density estimates are needed for the threatened Alaska-breeding population of Steller's eiders (*Polysticta stelleri*) to assess recovery status and guide recovery actions but these quantities are poorly estimated in this rare species using traditional methods. Therefore, population size and spatial variation in density were estimated across the Arctic Coastal Plain (ACP) and Utqiagvik Triangle (Triangle) survey areas using spatio-temporal generalized additive models and compared to traditional design-based estimates. Compared to design-based estimates, model-based estimates were more precise and less variable between years. Moreover, data sets can be combined to produce common estimates, detection probability can be incorporated to estimate population size, and spatial density maps can be produced from model-based estimates. Across both data sets and when combined, Steller's eider populations fluctuated with an approximate 6.53-year period (95% CI: 5.46, 7.93) with populations cycling from modeled posterior lows of 25–500 individuals and highs from 230 to > 2000 individuals across the ACP. Over the long term, the posterior 25-year geometric mean growth rate was −0.02 (95% CI: −0.07, 0.02), and shorter term growth rates were less well-estimated and fluctuated between positive and near zero. Density maps showed a high concentration in the northern part of the Triangle area and lower densities southward and across the ACP. Models fit to just the Triangle area indicated that eider density has been shifting northward though time, but this pattern was not supported in the sparse ACP data set, nor was it well-estimated when the ACP and Triangle data sets were combined. Given the strong cycle in this population, estimates of population size at similar cycle phases should be used for trend estimates.

or otherwise used by anyone for any lawful purpose. The work is made available under the Creative Commons CC0 public domain dedication.

**Data availability statement:** Arctic Coastal Plain data from 2007 to 2024 used in this paper is available at https://doi.org/10.7944/qtgn-y170. A description of quality control processes and data manipulations are available at https://github.com/USFWS/ACP-Mapping. Data for the Triangle Survey are available at https://doi.org/10.7944/0x1v-4b77. Quality control process for these data and all code use for analyses, manuscript preparation, and figure generation are available at https://github.com/USFWS/STEI-estimates. Results from the analyses including R model objects, posteriors samples, and csv files summarizing various posteriors can be found at https://doi.org/10.7944/22x5-t856.

**Funding:** The author(s) received no specific funding for this work.

**Competing interests:** The author has declared that no competing interests exist.

## Introduction

The Pacific population of Steller's eider (*Polysticta stelleri*) is a sea duck that nests in northeastern Russia and Alaska, with the Alaska breeding population currently listed as Threatened under the Endangered Species Act. As such, frequent status assessments are needed to determine correct listing status and to guide recovery actions. Unfortunately, the rarity and low density of this eider in Alaska make inferences of population size, trend, and distribution especially difficult. Moreover, observations of Steller's eider on surveys are extremely variable among years, with some years having no observed eiders [1,2]. It has been hypothesized that these eiders forgo breeding in some years when brown lemming (*Lemmus trimucronatus*) abundance and their avian predators are low [3]. Such cycles or intermittent breeding would further complicate methods to estimate long term trends or model population viability [4]. The effort described here is an attempt to make the most of the available data using modern model-based methods to distinguish sampling variation from the underlying signal of population differences between years and locations.

The approach here is inspired by Miller et al [5] and Amundson et al. [6], where generalized additive models [7,8] are used to estimate temporal and spatial variation in eider density. This approach to estimating population trend also borrows from the work of [9–11] among others, although the focus here is only one species. In general, the advantage of model-based estimates are that they help to separate noise (i.e., sampling variation) from the true underlying signal (i.e., temporal or spatial patterns in density [12,13], many others) and is closely related to the concept of shrinkage of random effects [12,14], where information is shared across samples in time, space, or other groups. This contrasts with pure design-based estimates, which make minimal assumptions, but suffer from high variability in this rare species. Specifically, sampling variation alone might lead to observations of zero eiders even when eiders did exist in the survey area. While correcting for the detection process is possible for design-based estimates, in the case of zero observations, no correction can be made, and these years will tend to under represent the true animal density. Sampling variation also inflates the year-to-year variance in estimates and will tend to make population trend estimates less precise and potentially more extreme, especially when multiple species are compared or statistical significance tests are used to detect trends [9]. In contrast, model-based estimates require the specification of a probability model for observations and help distinguish sampling from non-sampling variation. In the spatial and temporal context, where correlation is generally high and positive between nearby measurements, models that exploit this feature are especially helpful, and nearly all modern methods to detect trends in time and space use such models (see citations above, and [15,16]).

Below, I apply spatio-temporal models to provide the best available estimates of Steller's eider density and annual population size on the Arctic Coastal Plain (hereafter, ACP) and a smaller but highly sampled triangle-shaped area (Triangle survey) near Utqiagvik, Alaska. I first describe the data (section "Survey areas and data source"). I then describe the calculations of design-based estimates (section "Design-based estimates"). This is the first report of design-based estimates for the

Triangle data that include an estimate of uncertainty. These provide a reference point against model-based estimates. I then describe the data formatting and a variety of spatio-temporal generalized additive models fit to the data (section "Model-based estimates"). Posterior simulations from the fitted model estimates are then used to calculate total population size and trend (section "Prediction and posterior simulation"). I also describe the detection estimate used during posterior simulations (section "Incorporating detection"). Finally (section "Wavelet analysis"), I conduct a wavelet analysis of the posterior mean population size across years to estimate and statistically test for a periodic cycle in the eider population. This is of practical significance because it has been hypothesized that nesting effort of Steller's eiders is highest when brown lemming populations are at or near peak of their boom-bust cycles, and if there is a strong cycle, estimates of trend will depend on the relative endpoints used in building the estimate. Thus, if the population is cyclic, consistent relative positions in the cycle should be used for trend estimates. Throughout the results I provide some interpretation and provide comparisons to past work (section "Results and Interpretation"). I then provide some recommendations for model-based estimation of population size and trends, compare these methods to related efforts, and discuss the importance of the population cycles and detection estimates for species status assessment under the US Endangered Species Act (section "Discussion", section "Conclusion").

## Methods

### Survey areas and data source

Data used in this report came from the ACP Survey conducted by the United States Fish and Wildlife Service, Division of Migratory Bird Management, Alaska Region, and a survey conducted by ABR, Inc.- Environmental Research and Services (ABR) near Utqiagvik, Alaska, under contract by the United States Fish and Wildlife Service, Ecological Services Field Office, Fairbanks, Alaska, (hereafter, the Triangle survey, Fig 1). The ACP survey has been described in [6] and [2]. Unlike in [6], here only data from the actual ACP survey (2007–2019 and 2022–2024) are used. Amundson et al. [6] also included data from a different survey (before 2007) with different seasonal timing that causes a difference in observed response (a major subject of the effort in [6]). Data collected before 2007 have also not undergone the same level of quality control, which was a major effort for this analysis. The ACP data from 2007 to 2024 used in this report is available at

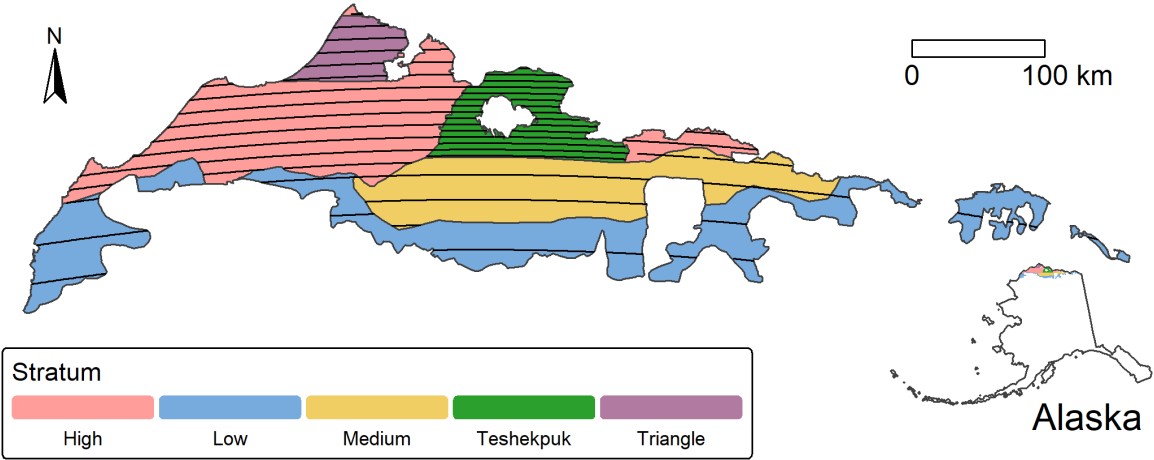

**Fig 1. Arctic Coastal Plain (ACP) and Triangle survey areas.** East-west transect lines are shown in black for one year on the ACP. Strata with different transect sampling intensities for the ACP are shown by different fill colors. The Triangle area is shown as a stratum, but it is completely contained within the ACP High stratum. Transects for the Triangle survey are very close (800 m in most years) so would appear as a solid color at this scale. On the Arctic Coastal Plain survey, transects are rotated north-south over a four year period to increase spatial coverage (not shown). All map data are products of the U.S. Government and are in the Public Domain.

. A description of quality control processes and data manipulations are available at https://github.com/USFWS/ACP-Mapping.

The Triangle survey is described in reports from ABR [1,17]. Data for the Triangle survey were obtained from ABR and are available at https://doi.org/10.7944/0x1v-4b77. A lengthy quality control process was applied to make data available for use in this report (documented in the file `wrangle_ABR.R`, available at https://github.com/USFWS/STEI-estimates).

Important differences between the surveys include the much higher sampling coverage of the Triangle survey (25–50%) compared to the ACP survey (approximately 1–8%), the much larger area of the ACP survey, and the sampling effort is stratified in the ACP survey (Fig 1). Hen eiders, birds occurring off of defined transects, and behavior (flying/not flying) are also recorded on the Triangle survey, but these are not generally recorded during the ACP survey. In addition, the ACP survey records all waterbird species, but the Triangle survey records a smaller subset (king [*Somateria spectabilis*] and spectacled [*Somateria fischeri*] eiders in addition to Steller's, other species in some years). Otherwise, the two surveys follow standard aerial waterfowl survey protocols [18]. No flocked Steller's eiders were observed on either survey, and females outside of male-female pairs and all off-transect observations were dropped from the Triangle data for all analyses. There was one case of an "open 1" Steller's eider in the ACP data set (a general category for a group of unknown sex or pair status that is not appropriate for an observation of a single), and it was assumed to be a single male.

All general data manipulation and analysis was done with program R [19] in RStudio [20] with the tidyverse packages [21]. An initial version of the manuscript was produced using Quarto (https://quarto.org/) and the PLOS One journal template found at https://github.com/quarto-journals/plos in an effort to maximize reproducibility. Code for these analyses, results, and production of this manuscript is maintained at https://github.com/USFWS/STEI-estimates.

**Design-based estimates**

I calculated a design-based estimate using formula R3 of Fewster et al. [22] (also see [23], p. 79) modified for strip transects. A ratio estimator, which is more common for strip-transect surveys [24–26], was investigated but found to be poor (very high variance) because most observations of Steller's eiders are on short transects in the northern end of the triangular-shaped area and longer transects in the south rarely have observed eiders. The ratio estimator can provide lower variance estimates when there is a positive correlation between the two variables (here, animal count and transect length or area) [24,26]. In this survey area, however, there is a negative correlation between count and transect length (or area). This makes the ratio estimator worse than a simple plot-based estimator or the R3 estimator of Fewster [22]. The point estimate was calculated as

$$\hat{N} = A \frac{\Sigma_i n_i}{\Sigma_i a_i}$$

(1)

and the variance of $\hat{N}$ was calculated as

$$var(\hat{N}) = \left(\frac{A}{a}\right)^2 \frac{a}{k-1} \sum_i a_i \left(\frac{n_i}{a_i} - \frac{n}{a}\right)^2$$

(2)

where $A$ is the total area of the study area, $a_i$ is the area of strip transect $i$, $n_i$ is the number of encountered single males or pairs on strip transect $i$, and $n$ is the total encounters on the $k$ surveyed strips, and $a$ is the total area of the $k$ surveyed strips. It is arguable if a finite population correction, $(1 - a/A)$, is appropriate, so I left it out as it is not used by Fewster et al. [22] or Buckland et al. [23] because observations on transects are not repeatable due to movement of birds and the detection process. Fewster et al. [22] showed that Eq (2) can overestimate the variance of systematic surveys when there is a strong gradient in density, as there is in this area. In this specific case, however, the estimator above is much better than a standard ratio estimator.

## Model-based estimates

I used generalized additive models (GAMs, [7,8,27]) to estimate eider density as a function of location and time. A GAM can be thought of as a generalized linear mixed model that fits smooth functions (splines) of covariates to predict the response, in addition to standard linear mixed model terms if specified. The optimal degree of smoothing is determined during model fitting through a model selection-like process. The smoothed temporal or spatial effect, thus, can account for autocorrelation in these domains [8,27,28]. GAMs are widely used in animal density surface modeling (e.g., [5,6,29] and many others).

To set up the data for model fitting, I divided sampled transects into 1 (Triangle survey) or 6 (ACP survey) km segments and assigned observations of eiders (including observation of zero eiders) to segment centroids. Segments on the boundary of the study area were often smaller due to boundary issues. I then assumed a half-strip width of 200 m and calculated an area for each segment. Total number of eider observations (pairs and males) were summed for each segment and the coordinates (in Alaska Albers Equal Area projection, EPSG:3338) of the centroid were used for spatial location. Year was used as the covariate for temporal effects. The same procedure was used for both the entire ACP and for the Triangle survey, but for the ACP-only model (see below) the segment size for model fitting was 6 km. This was done for computational reasons (time) related to fitting many different models and to be consistent with [6]. When Triangle and ACP data were combined into one model, a common 1 km segment size was used. All spatial data manipulation was done using the R package `sf` [30].

I fit a variety of models to explore spatial and temporal effects. The linear predictor for each model was:

$$M0 : Count \sim s(X, Y)$$

$$M1 : Count \sim s(X, Y) + s(Year)$$

$$M2 : Count \sim s(X, Y) + s(Year) + ti(X, Y, Year)$$

$$M3 : Count \sim s(X, Y) + s(Year) + s_{re}(fYear)$$

$$M4 : Count \sim s(X, Y) + s_{re}(fYear)$$

In the above, *Count* is the total number of pairs or males observed in a transect segment, $s()$ indicates a smooth function, $ti()$ indicates a "tensor product smooth" (a multidimensional smooth that allows the units of the dimensions to differ), and $s_{re}()$ is a random effect in the usual mixed model sense (in `mgcv: s(..., bs=re)`). *Year* is calendar year as a continuous numeric variable, and *fYear* is calendar year as a factor variable. Model *M0* is just a smooth of location that does not change through time. Model *M1* is a smooth of location and a separate smooth of year. This model means that on the log scale the spatial smooth does not change its relative shape through time but the overall height changes as a smooth function of year. Model *M2* is a smooth of location, year, and a spatio-temporal interaction between location and year. Model *M2* allows the shape of the spatial pattern to change through time and allows the time trend to depend on location. Model *M3* estimates a smooth of location, a smooth of year, and a separate random effect of year, which allows abrupt non-smooth deviations from an underlying smooth trend [11]. Model *M4* estimates a smooth of location and treats year as a simple random effect. The model with year as a random effect still imposes smoothing on the year effect, just as in the usual mixed model, where year effects are pulled or "shrunk" toward an overall grand mean. In the models above, each effect is written as an "average" or "partial" effect relative to the others and all contain an intercept (not shown). Thus,

*s*(*X*, *Y*) + *s*(*Year*) is an average effect of location and an average effect of Year as a deviation from an overall intercept. All models used a negative binomial response, a log link function, and the log segment area as an offset, which controlled for varying areas of segments near study area boundaries. Thus, the model is estimating the expected response in 1 *km*² of area. The scale parameter of the negative binomial was estimated during model fitting. Other response models were fit (Poisson, Tweedie) and all were found to be substantially worse fitting than the negative binomial. I used a 'thin plate regression spline' (s(..., bs="tp") in mgcv, [31]) as the spline basis function for all smooth effects in all models. All model fitting and prediction was done using the R package mgcv [31]. Model diagnostics were inspected using the residual simulation methods in R package DHARMa [32] and visualizations using the R package gratia [33]. Special attention was given to quantile-quantile plots and over-dispersion and zero-inflation metrics. Residual simulations suggested that the negative binomial distribution appropriately modeled the large number of response zeros in these data. To select the best model, the AIC function was used with the correct degrees of freedom for a GAM (see ?mgcv::AIC.gam in [31]).

For modelling the ACP data alone, additional models were fit that contained a random effect for observer. The response for these models was observer-specific; therefore, each segment was only 200 m wide (i.e., each side of the plane was a separate observation 200 m wide). An additional five models were fit where each model above also contained an observer effect, $s_{re}$(*Observer*). Observer effects were not estimated for the Triangle data because they were not recorded in most years. When I combined the Triangle data with the ACP data, I used a fixed factor effect for survey (Triangle or ACP) to model any average difference between surveys and used a model with spatial and temporal smooths as in *M*1. A model with a spatio-temporal interaction (*M*2) was also explored for the combined Triangle and ACP data, but rejected. The two data series do not overlap during the period of 1999–2006, so the interaction term was not well estimated and produced widely variable population estimates during posterior simulations for the years 1999–2006. During posterior simulation (see below), the survey covariate was set to predict the Triangle survey because this is the area where detection was estimated. R code giving the details of model fitting can be found at https://github.com/USFWS/STEI-estimates.

## Prediction and posterior simulation

I used the predict.gam function from mgcv package [8,31] to predict the expected density of eiders across the study area (Triangle or ACP) in each year, from 1999 to 2023 for the Triangle and from 2007 to 2024 for the ACP. Note that years with no data collection are included in these predictions. To do this, the study area was gridded into 1 *km*² or smaller cells–smaller when the cells intersect the boundary of the study area–and centroids and areas of each cell were calculated. A data set was then created by replicating these point locations and areas for each year. This large data set was then used in the predict function along with the model object from the best fitting GAM model. For spatial maps to represent relative differences in eider density, predictions were made on the response scale (predict option type="response") and the year effect was excluded (exclude="s(Year)" or exclude=c("s(Year)," "ti(X-,Y,Year)")). Predictions at cell centroids were used for the entire cell, that is, the continuous smooth density surface was rasterized into 1 *km*² or smaller cells for display in maps.

Results of fitting a GAM in mgcv provide a posterior distribution of model parameters [8,34]. Therefore, posterior simulation was used to calculate population totals over the study area for each year (see examples in help files for mgcv::predict in [31] or [8], p. 342–343). Predictions were made once on the same grid and years as described above but type="lpmatrix" was specified so that a design matrix, **X**, was returned with one row for each prediction location-year and one column for each term in the model. Multivariate normal samples of the model parameters, **b**$_i$ were then simulated using the fitted model parameter vector and variance-covariance matrix. For the Triangle study area, direct simulation from a multivariate normal distribution was used. For all simulations on the ACP study area, a Metropolis-Hastings algorithm was used to obtain samples because large areas of zero observations caused poor performance based on direct multivariate normal simulation using the parameter estimates and covariance matrix (see ?mgcv::gam.mh in [31] or [35]). A posterior sample on the response scale was then calculated as

$$\mathbf{y}_i = \mathbf{a}exp(\mathbf{Xb}_i)$$

(3)

where $a$ is a vector of the area of each grid cell and $\mathbf{y}_i$ is a vector of the expected responses. Note that $\mathbf{y}_i$ is the expectation of the negative binomial distribution and not an actual realization; thus, it is > 0 for all predictions. The above simulation was repeated a large number of times (500) and results were stored.

Various derived quantities of the posterior samples can also be calculated. To find the expected population total in a given year, the sampled vector can be summed over all cells for a given year. Let $i$ index the posterior sample, $j$ index year, and $k$ index the cell, then a posterior sample for the expected population total in year $j$ is

$$\hat{Y}_{ij} = \sum_k y_{ijk}.$$

(4)

Because the model was fit to pairs and single males, the above gives the total "indicated pairs." To transform this to birds and "indicated birds," the above posterior sample would be multiplied by 2. A detection corrected population total for "indicated birds" can be found as

$$\hat{N}_{ij} = 2\hat{Y}_{ij}/d_{ij}$$

(5)

if detection varies only by year, where $d_{ij}$ is a posterior sample of detection in year $j$. If detection varies with location or other covariates, then the adjustment needs to take place at a lower level of $y_{ijk}$.

The posterior trend on the log scale from year $j$ to $j + t$ can be found as

$$T_{it} = log\left[(\hat{N}_{ij+t}/\hat{N}_{ij})^{1/t}\right] = \frac{log(\hat{N}_{ij+t}) - log(\hat{N}_{ij})}{t}.$$

(6)

Note that this measure of trend is identical to the slope parameter in a "log-linear" regression when the smooth of year in the GAM model, $s(Year)$, gives a straight line. The advantage of the GAM is that the notion of trend can be generalized to non-linear cases where the trend may be increasing, decreasing, or changing cyclically through time. The posterior distribution for any quantity can then be displayed using a histogram or summarized with various metrics. I summarized the posterior with the mean and the 0.025, 0.5 (median), and 0.975 quantiles. For trend, I only used detection corrected posterior estimates (see below). Full detail of the posterior simulations can be found at https://github.com/USFWS/STEI-estimates.

**Incorporating detection**

I used a preliminary detection rate estimated from the second year (2018) of an on-going field study using artificial decoys to estimate detection. The detection analysis was completed by Catherine Bradley (USFWS, retired) and showed that detection rate in 2018 depended only on distance from transect (see S1 Table). Detection was also estimated during 2017, the first year of the study, but field protocol issues caused problems during this pilot year, and the protocol was stabilized in 2018. Because distance was not available for observations outside the decoy detection study area (a sub-set of the Triangle area), I calculated the average 'unconditional' detection rate over the transect half-strip width (S1 Table). In this context, 'unconditional' is the detection rate estimated from double-observer sightability trials that include decoys not observed by either observer (a '00' capture history). I also used this same detection rate for the ACP because no better estimate for Steller's eider is available. The detection rate used here should be viewed as provisional until a more complete analysis can be conducted incorporating data from across multiple years and crews. In any case, it is meant to be the average detection rate over all observations, including

the covariates of distance, year, observer, sun angle, etc., and was in fact used for the Steller's eider Species Status Assessment of 2025 [36].

To calculate an average detection rate, I assumed that eiders (detected and undetected) were expected to be uniformly distributed with distance from the transect, which is true given the design of the Triangle and ACP surveys. I then averaged detection and the variance of the detection estimate over each of the four distance bins in S1 Table. Because distance bins were of equal width, detection was estimated as a factor of bin (i.e., a continuous distance function was not estimated), and no other covariates were found important, the mean and variance are a simple equally weighted average over the distance bins. In S1 Table only the mean and ninety-five percent credible interval was given, so I approximated the standard error of the estimate in each bin by the range divided by 2*1.96, which is the typical number of standard deviations in the range of a symmetric credible interval.

This worked out to a detection rate of 0.307 with a standard deviation of 0.092 or a Beta(7.41, 16.72). Note that this detection prior will result in a large amount of uncertainty in the estimated number of eiders. At detection rates of 0.16 and 0.47 (the fifth and ninety-fifth percentile of the distribution, respectively), the eider population estimate will be increased about 6- and 2-fold, respectively. Thus, increased information on detection would be valuable for improving our knowledge of the eider population size. Interestingly, this detection rate prior has a very similar mean but is much more variable compared to that used in the past based on expert judgment [4].

The posterior population estimate for "indicated breeding birds" (single males and pairs), accounting for constant detection, was then calculated as

$$\hat{N}_{ij} = 2\hat{Y}_{ij}/d_i. \tag{7}$$

with $d_i$ sampled from a Beta distribution with the mean and standard deviation above. Thus, detection was assumed constant across years and other covariates (e.g., crews, singles, pairs). If detection varies with year, location or other covariates, then the adjustment needs to take place at a lower level as in (Eq 5).

In general, quantities $\hat{Y}$ are designated as "indicated pairs" because single males are assumed to "indicate" an observed female on the nest (the male-female sex ratio is assumed equal). (Technically, within the tradition of waterfowl biology, "indicated pairs" include flocked drakes in groups of less than 5. However, none of these were observed in either data set; thus, "indicated breeding pairs" and "indicated pairs" are the same.) When "indicated pairs" are multiplied by 2, $2\hat{Y}$, then the quantity becomes "indicated birds" because each pair is 2 individuals. Hereafter, I use "indicated breeding birds," and I use the word "index" to distinguish quantities that are not corrected for detection. Thus, $2\hat{Y}$ is the "indicated breeding bird index," and $2\hat{Y}/d$ is simply "indicated breeding birds," to reflect that it is a population estimate, at least conditional on the assumptions inherent with "indicated."

## Wavelet analysis

In order to formally estimate the period of a repeating cycle in the Steller's eider population, I used wavelet analysis to detect and statistically test for the presence of a cycle. Wavelet analysis is similar to a Fourier transform of a time series where the time domain is expressed in the frequency domain. Unlike a basic Fourier transform, however, a wavelet transform localizes the frequency information in time so that changes in the frequency of a signal with time can be examined. The main product of a wavelet analysis is a "power spectrum" (also know as a spectrogram) that is a three dimensional surface showing the "power" (z-axis or a measure of the strength of a particular frequency or period) in a time series as a function of the wave period (y-axis) and time (x-axis). High "power" ridges across time correspond to a large signal at that frequency (or period) in the time series. If the period is changing through time, the position of the ridge will change relative to the period axis as time increases. An accessible introduction can be found in [37] and in the documentation to the R package `WaveletComp` [38]. I used the function `analyze.wavelet`

from the `WaveletComp` package [38] to calculate the wavelet properties of the posterior mean population time series estimate from the model that combined both Triangle and ACP data. I used the null hypothesis of white noise and 200 bootstrap simulations to test for statistical departure form white noise (a flat or uniform frequency spectrum) and to calculate a confidence interval for the period. I then used the functions `wt.image` to visualize the power spectrum and `wt.avg` to visualize the average power spectrum across years. To find the full posterior of the dominant wave frequency, I calculated the wavelet properties of 500 posterior samples of the population time series from the best fitting GAM model and saved the wave period at the maximum average power across time for each sample. I then used the mean and 95% credible interval to summarize this posterior.

## Results and Interpretation

### Observations of Steller's eider

Observed locations of Steller's eiders across the study area are shown in Fig 2. Most observations are in the northern coastal area of the Triangle and Teshekpuk area. No observations of Steller's eiders have been made in the two southern-most ACP strata. Observations from the Triangle survey abruptly stop at the southern edge of the Triangle survey area, suggesting that eiders sometimes exist south of that area more often than observations from either survey suggest. Fig 3 shows the count of Steller's eiders by year for both the Triangle and ACP surveys.

### Design-based estimates

Design-based estimates for the both survey areas are shown in Fig 4. No survey was conducted in 2020 for the Triangle survey or in 2020 and 2021 for the ACP. Note the higher mean and variance of estimates for the ACP compared to the Triangle survey and that estimates for the ACP often overlap zero. This is due to the larger area and smaller sampling fraction of the ACP design. Design-based estimates make minimal assumptions and provide an important check on model-based estimates. Two important assumptions behind design-based estimates are that (1) the survey design is unbiased (random or systematic selection of sampling units, here transects) and (2) the survey was implemented as designed. These assumptions are met in these surveys. An additional often unstated assumption is that the response is measured without error. In this case, measurement errors include a large detection bias and, presumably less often, species misidentification. Detection bias causes a lower mean response and increased variance in response. Finally, design-based estimates are based on estimating a mean response across all sampled transects, and then use

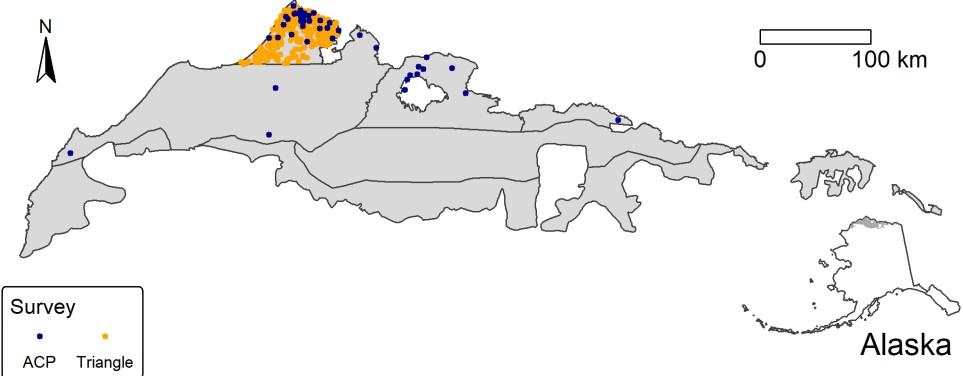

**Fig 2. Observed Steller's eider during the Arctic Coastal Plain (2007-2024) and Triangle (1999-2023) surveys.** Arctic Coastal Plain (ACP) strata are shown by black lines. Locations of eider observations by survey are given by the purple (ACP) or orange (Triangle) dots. All map data are products of the U.S. Government and are in the Public Domain.

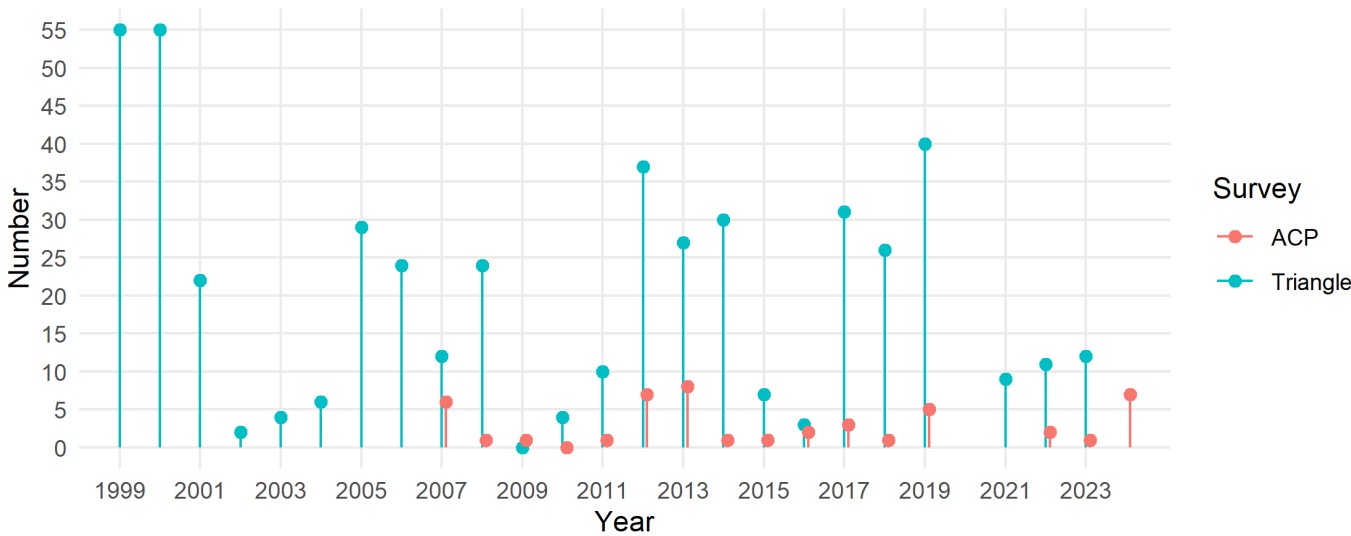

**Fig 3. Number of Steller's eider observations by year for the Arctic Coastal Plain and Triangle surveys.**

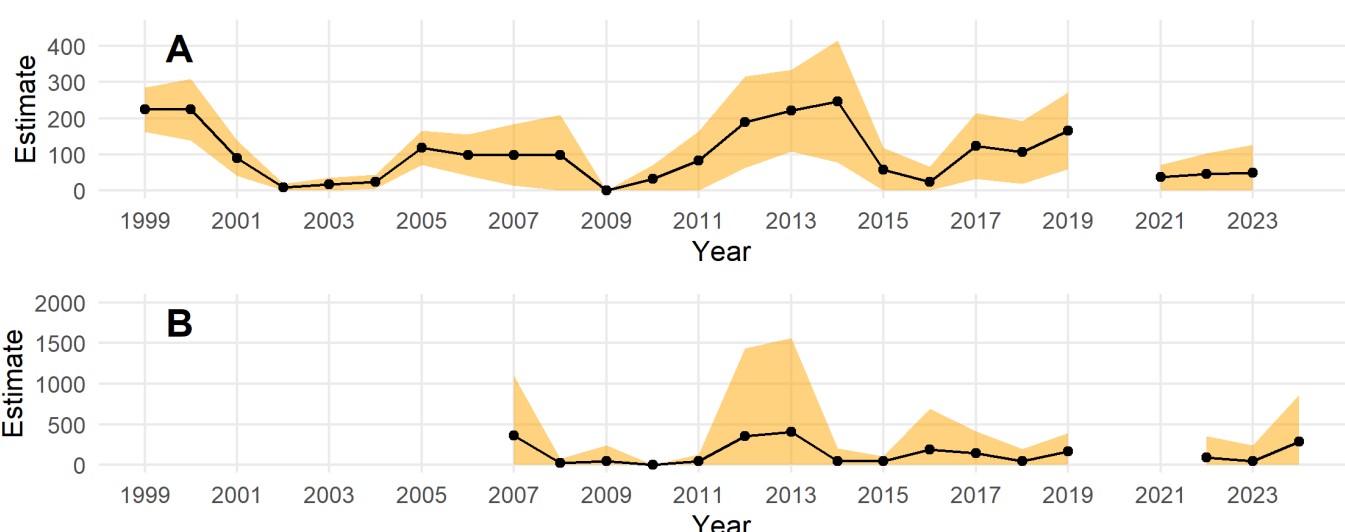

**Fig 4. Design-based estimates for each survey.** (**A**) Triangle survey area and (**B**) Arctic Coastal Plain survey area. Point estimates for each year are given by black dots connected by black lines across years. Confidence intervals (95%, orange band) have been truncated at zero. No detection correction was applied.

statistical sampling theory to derive estimator variance. This works well for common species and large sample size (many transects), but for rare species that might not be encountered on any transects during a sample, the estimate and its variance are necessarily zero when there are no detection events, even when the species might have existed in the survey area. This might have been the case in 2009 on the Triangle area (Fig 4A) and in 2010 for all strata of the ACP (Fig 4B), and was the case in 2, 10, 16 and 16 years out of 16 for the High, Teshekpuk, Medium, and Low strata of the ACP, respectively.

## Model-based estimates: Triangle

Models fit to data from the Triangle showed that the best model contains a spatial smooth, a year smooth, and a spatio-temporal smooth (model M2, Table 1). The model with a separate space and time smooth (M1) and all other models were worse ($\Delta AIC > 5$). For all results below, I used model M2.

The spatial, temporal, and spatio-temporal partial effects of model M2 are shown in Fig 5. The largest approximate p-value for the smooth terms was 0.003 for the spatio-temporal smooth. Effective degrees of freedom for the spatial (Fig 5A), temporal (Fig 5B), and spatio-temporal (Fig 5C) smooth were 16.1, 12.1, and 2.1, respectively. Highest densities of Steller's eiders were in the northern section of the Triangle and the lowest densities were in the southeast, but as shown by the spatio-temporal smooth (Fig 5C), density decreased through time in the southeast and increased in the northwest of the Triangle. Note that the spatio-temporal smooth has essentially been shrunken to a simple 2D plane that is changing through time such that eider density is increasing in the northwest and decreasing in the southeast. Full `mgcv` model objects and parameter values can be found at https://doi.org/10.7944/22x5-t856. The temporal pattern appears cyclic with a period of 5–7 years and no strong directional trend (Fig 5B).

Spatial density of Steller's eider across the Triangle after removing the effects of year and the space-by-year interaction (i.e., the average or partial effect of location) is shown in Fig 6. Relatively high densities have occurred in the north and moderate densities in the east and far southwest. With the spatio-temporal effect, however, these areas of higher densities in the south have decreased.

Population estimates across the Triangle survey area by year for indicated breeding birds without (Fig 7A) and with (Fig 7B) a detection correction are shown in Fig 7. There is a substantial increase in the population estimates, the uncertainty, and in the skew of the posterior toward higher population estimates with the application of the a detection correction (Fig 7B). This is due to the low mean detection rate and high uncertainty in the detection prior. Note that the model predicts the population in 2020 when no survey was conducted. The population appears to cycle on a period of 5–7 years (Fig 7B); thus, causing the posterior n-year trend to fluctuate at the posterior mean (Fig 8). Trend estimates are more precise for longer lags, but also fluctuate from positive to negative at the posterior mean (Fig 8).

## Model-based estimates: ACP

Models fit to the ACP data showed that a model with a random effect of year, rather than a smooth of year, fit best (Table 2). In addition, models with a random observer effect contributed nothing to improvement in AIC (Table 2). Upon examining model summary statistics, models with a smooth of year shrunk to a straight line and observer effect variance was essentially zero (results can be found in model objects of https://doi.org/10.7944/22x5-t856). These results are likely due to the few observations of Steller's eider on the ACP survey and the overall sparse nature of the sampling. The year random effect of the best model had the largest p-value (0.008) and an effective degrees of freedom of 7.9, indicating substantial

**Table 1. AIC table for models fit to Triangle data.**

| Model | Linear Predictor | df | AIC | ∆AIC |
|---|---|---|---|---|
| M2 | s(X,Y) + s(Year) + ti(X,Y,Year) | 35.48 | 2876.13 | 0.00 |
| M1 | s(X,Y) + s(Year) | 33.63 | 2881.94 | 5.81 |
| M3 | s(X,Y) + s(Year) + s(fYear) | 39.89 | 2884.29 | 8.15 |
| M4 | s(X,Y) + s(fYear) | 38.18 | 2888.06 | 11.92 |
| M0 | s(X,Y) | 19.78 | 2979.11 | 102.98 |

Model structures are described in the main text.

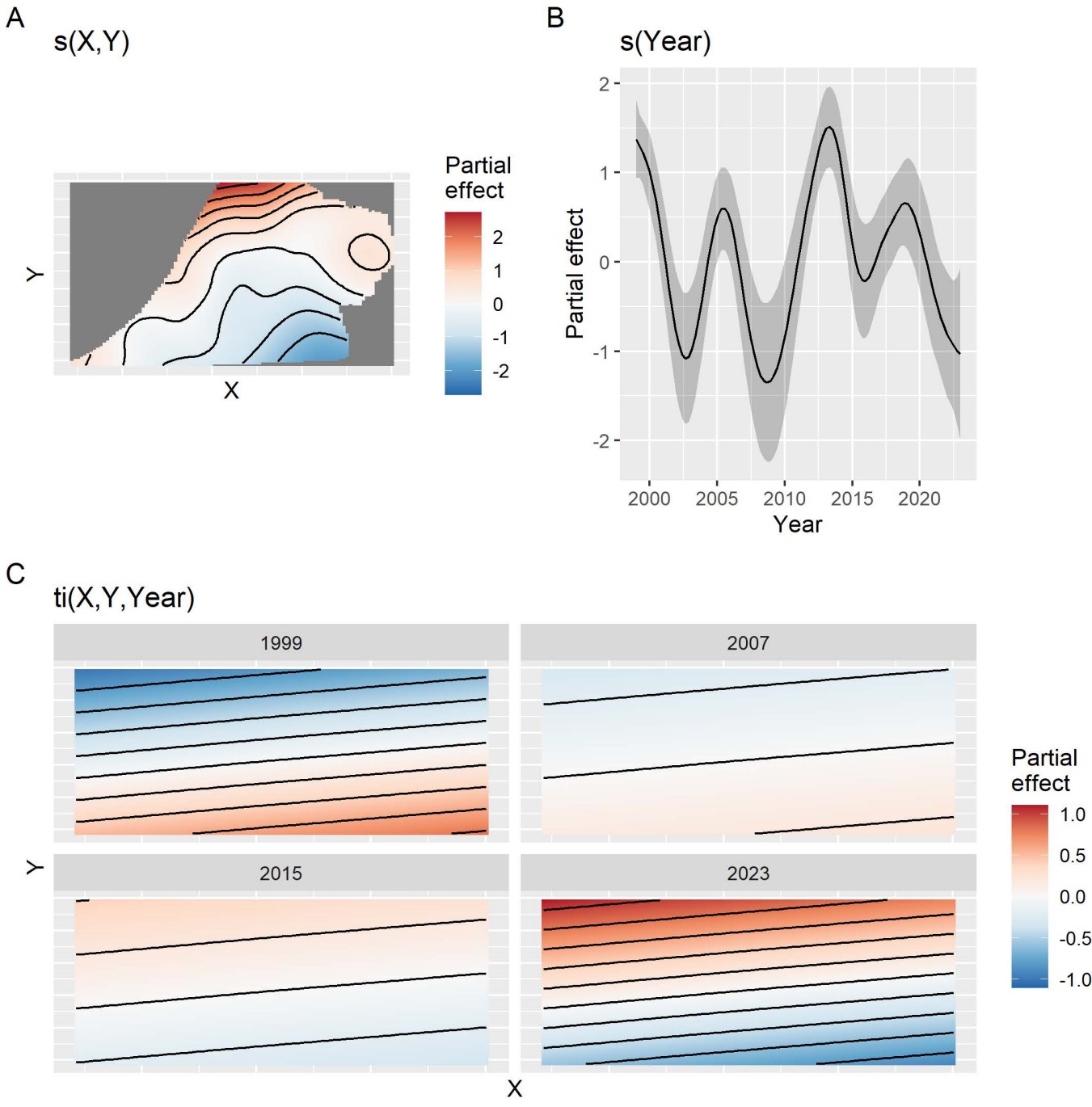

**Fig 5. Partial effect plots for the best model fit to the Triangle survey data.** Plots of spatial **(A)**, temporal **(B)**, and spatio-temporal interaction **(C)** partial effects from the best fitting model. In (B) the shaded region is 2 standard errors. In (A) and (C) black lines show contour lines of equal partial effect, and color shows increased (red) or decreased (blue) density relative to the intercept. In (C) the interaction is shown by 4 density surfaces at different time points from 1999 to 2023. Each panel is as in (A) but has been simplified to the bounding rectangle of the survey area. Note that as time progresses, the upper left of each panel increases in density as indicated by the shift from blue to red.

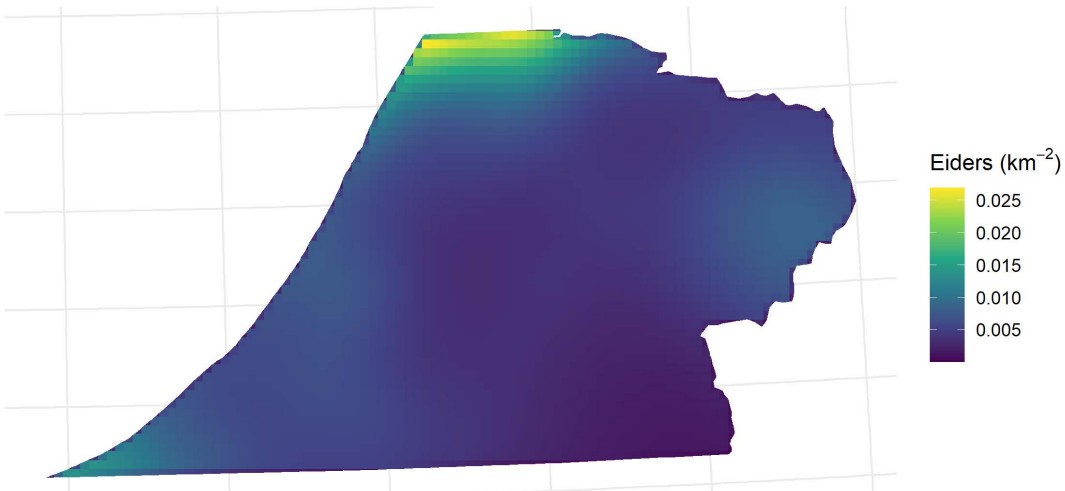

**Fig 6. Predicted average density of Steller's eider in the Triangle survey.** Predicted density after the effects of year and space-by-year interaction have been removed.

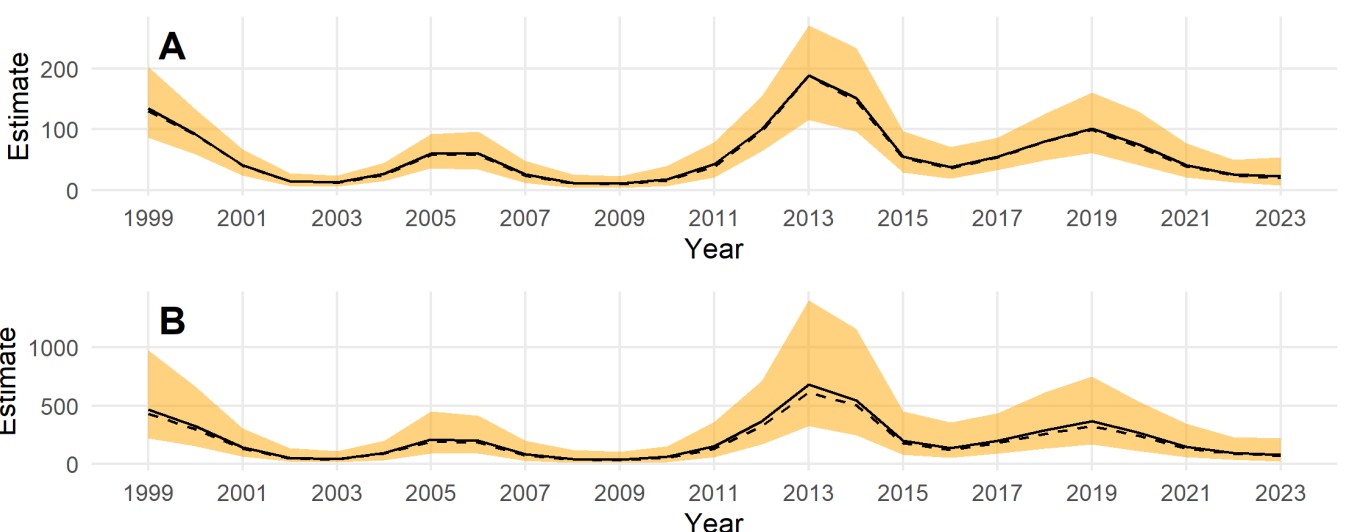

**Fig 7. Posterior estimates of Steller's eider in the Triangle survey area, 1999-2023. (A)** without accounting for detection and **(B)** after applying a detection correction. The black line is the posterior mean, the dashed black line is the posterior median, and the orange band is the 95% credible interval.

shrinkage toward the mean of the 16 year effects. The spatial smooth was substantially simpler (effective degrees of free-dom = 9.8) than the best model for the Triangle and showed a simple decline in eider density as distance from the coast increased (Fig 9). Despite the shrinkage of the temporal random effects, they still varied greatly (see below).

Posterior simulations of the year-specific total across the whole ACP are shown in Fig 10 without a detection correction (Fig 10A) and with a detection correction (Fig 10B). The population appears to fluctuate in a similar manner as in the model fit to Triangle survey data, but the year-specific estimates for the ACP are much less precise. Because the best model contained only a simple random effect to model year and the survey was not completed in 2020 and 2021,

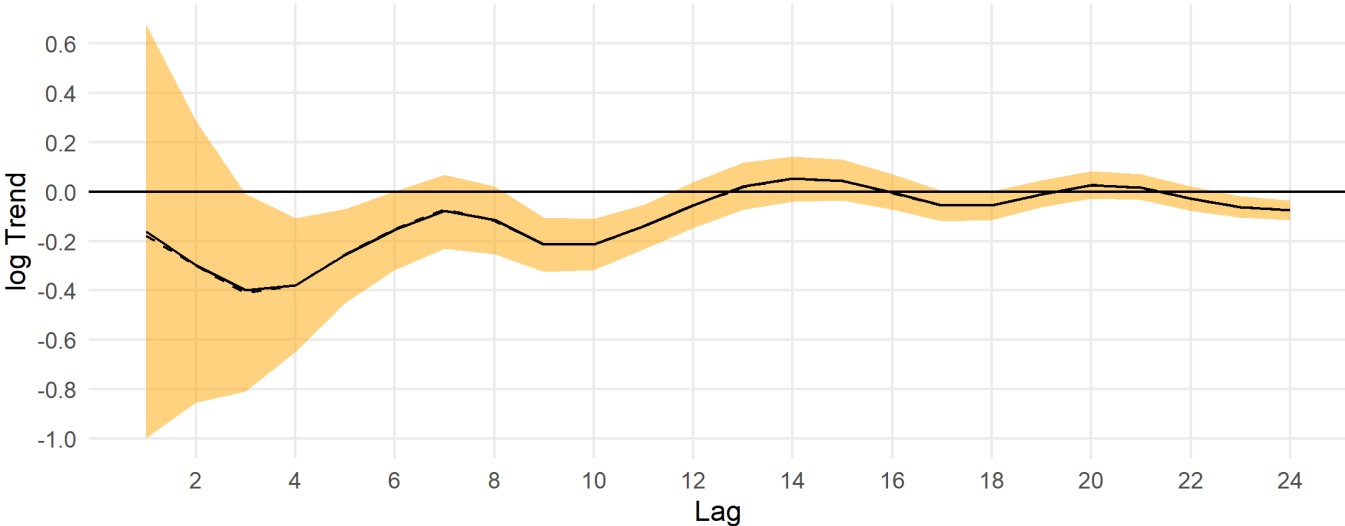

**Fig 8. Posterior trend estimates for Steller's eider in the Triangle survey area, 1999-2023.** The y axis is the log of the geometric mean growth rate, and the x axis is the lag-year trend, the lag of 10 gives the 10-year trend from 2013 to 2023. The black line is the posterior mean, the dashed black line is the posterior median, and the orange band is the 95% credible interval. The horizontal thick black line is the y-axis origin.

**Table 2. AIC table for model fit to Arctic Coastal Plain data.**

| Model | Linear Predictor | df | AIC | Δ ΔAIC |
|---|---|---|---|---|
| M4 | s(X,Y) + s(fYear) | 23.65 | 531.58 | 0.00 |
| M4.obs | s(X,Y) + s(fYear) + s(Observer) | 23.65 | 531.58 | 0.00 |
| M3 | s(X,Y) + s(Year) + s(fYear) | 24.51 | 531.83 | 0.25 |
| M3.obs | s(X,Y) + s(Year) + s(fYear) + s(Observer) | 24.53 | 531.86 | 0.28 |
| M0 | s(X,Y) | 14.54 | 537.44 | 5.86 |
| M0.obs | s(X,Y) + s(Observer) | 14.54 | 537.44 | 5.86 |
| M1 | s(X,Y) + s(Year) | 15.60 | 539.22 | 7.64 |
| M1.obs | s(X,Y) + s(Year) + s(Observer) | 15.60 | 539.22 | 7.64 |
| M2 | s(X,Y) + s(Year) +ti(X,Y,Year) | 14.07 | 540.47 | 8.89 |
| M2.obs | s(X,Y) + s(Year) + ti(X,Y,Year) + s(Observer) | 14.07 | 540.47 | 8.89 |

Model structures are described in the main text. The suffix '.obs' indicates a model that contained a random effect of observer.

predictions were not made for these years using this model. Such predictions could be made by simulating random effects for these years but this would simply give an estimate that spans the range of historical estimates because there is no temporal correlation in this model.

Note that year-specific estimates of the total number of eiders across the ACP based on model *M*4 (Fig 10) are less extreme than those based on design-based estimates (Fig 4B). That is, the model-based estimates are lower in high abundance years and higher in low abundance years compared to the design-based estimates. This is due to the 'shrinkage' or regularization of the year random effect. In general, design-based estimates might over-estimate the eider population when one or more eiders happen to be observed on a transect in one year and under-estimate when one or more eiders are not observed on a transect, even though they are in the study area.

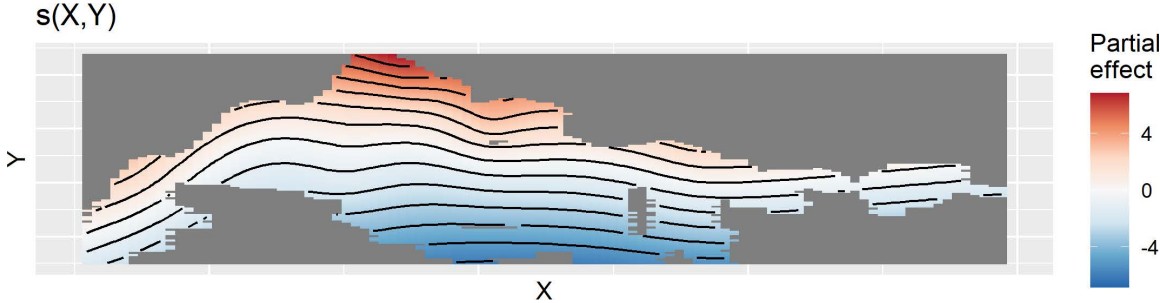

**Fig 9. Partial effect of the spatial smooth for the Arctic Coastal Plain model M4.** The map of density shows decreasing density further to the south or away from the coast. Black lines show contour lines of equal partial effect, and color shows increased (red) or decreased (blue) density relative to the intercept.

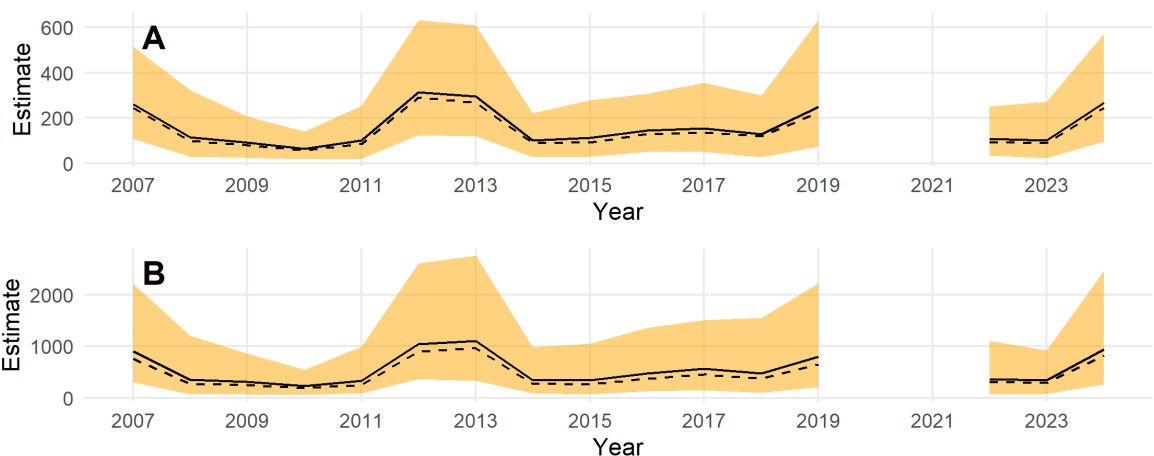

**Fig 10. Posterior estimates of Steller's eider across the Arctic Coastal Plain survey area, 2007-2024. (A)** without accounting for detection and **(B)** after applying a detection correction. The black line is the posterior mean, the dashed black line is the posterior median, and the orange band is the 95% credible interval.

Trends for the Steller's eider population across the ACP are shown in Fig 11. For all time ranges (lags), the trend was not well estimated. For example, the 10-year trend posterior mean was 0.11 (95% CI: −0.03, 0.23), which could be a modest decrease to a very fast increase (cf., Fig 10B). For all time periods, the 95% credible interval overlaps 0 and the upper bound was often > 0.15 (Fig 11).

## Model-based estimates: Triangle and ACP combined

Partial effects of the model fit to combined data are shown in Fig 12. This model contained a spatial smooth, a temporal smooth, and an effect of survey crew, but no spatio-temporal interaction. All p-values for the smooth terms were very small (< 0.001) and the survey crew effect p-value was 0.05. Effective degrees of freedom for the spatial and temporal smooths were 19.2 and 12.8, respectively, indicating less shrinkage than the ACP-only model and similar to the Triangle model. Similar to the ACP-only model, eider density decreased from north to south and with distance from the coast (Fig 12A). Similar to the Triangle model, the temporal trend fluctuated with a 5–7 year period (Fig 12B). The survey effect was estimated at 0.46 (SE = 0.23) on the log scale. Because the Triangle survey was defined as the baseline (intercept), this means that the ACP crew observed approximately sixty percent higher density at the posterior mean on average as

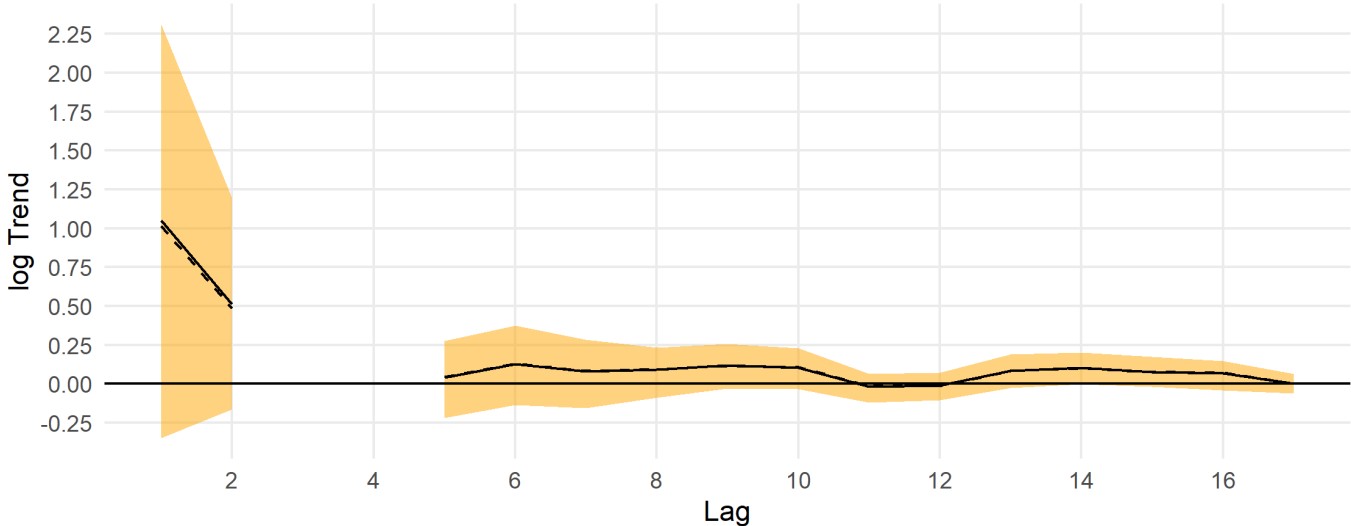

**Fig 11. Posterior trend estimates for Steller's eider in the Arctic Coastal Plain survey area, 2007-2024.** The y axis is the log of the geometric mean growth rate, and the x axis is the lag-year trend, the lag of 10 gives the 10-year trend from 2014 to 2024. The black line is the posterior mean, the dashed black line is the posterior median, and the orange band is the 95% credible interval. The horizontal thick black line is the y-axis origin.

compared to the Triangle crew after controlling for year and spatial location (95% CI: 1.7% − 147%). Spatial predictions on the response scale for 2013 are shown for the Triangle area in Fig 12C. Because there is no space-time interaction in the model, the relative density surface is a simple multiplicative relationship between all years as shown by the temporal partial effect (Fig 12B).

Posterior distributions for the year-specific population predictions over the entire ACP area are shown in Fig 13A without a detection correction and in Fig 13B after the detection correction is applied. Note that the predictions are made through 1999–2006 and 2021 when ACP survey data does not exist and for 2020 when no data exists. This is possible because of the space-time structure of the model. For 2020, predictions are due to the temporal correlation (smooth) and interpolating from nearby years when data exists. In 1999–2006 and 2021, the existing Triangle data is used to inform the temporal response, and then the temporally-constant spatial effect informs the prediction across space.

Posterior estimates of population trend show a strong increase over the most recent years and fluctuating trends as the time period increases (Fig 14). The posterior estimate for the 10-year trend is −0.03 (−0.15, 0.06) and for the entire series the 25-year trend is −0.02 (−0.07, 0.02).

## Wavelet analysis

A wavelet power spectrum clearly revealed a high power ridge across time that appears stationary with a period around 6 years (Fig 15). Another ridge of much lower power appeared at a 15 year period. When the spectrogram power is averaged across all years the maximum power was found at period 6.53 (95% CI: 5.46, 7.93). The lower power ridge gave a maximum average power at 15.26 years with a simulation pvalue of p=0.07, suggesting weak evidence for a longer period oscillation. When 500 posterior samples of the time series were sampled from the GAM and a spectrogram found for each sample, the posterior mean maximum power was found at period 6.52 (95% CI: 6.23, 6.76), suggesting very high confidence in maximum average power at this period across the range of posterior samples of the GAM.

Such periodic cycles will cause any estimate of trend to depend on the phase of the time points used for the calculation. Thus, when calculating trend, it might be useful to control for period so that start and end points reference the same relative position of the cycle. That is, when using the results presented here for trend, use approximate integer multiples

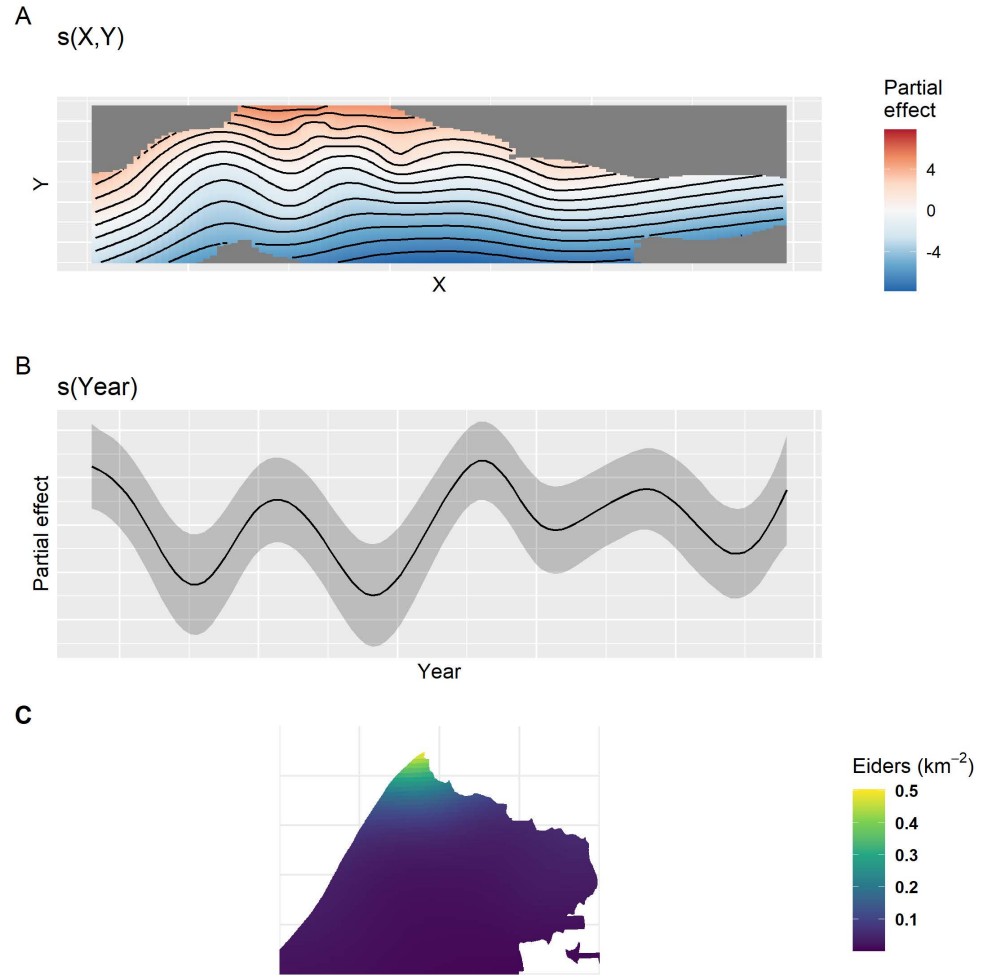

**Fig 12. Partial effect plots for the best model fit to the combined Triangle and Arctic Coastal Plain data.** Partial effect plots for **(A)** spatial and **(B)** temporal effects. There is no space-time interaction in this model. **(C)** spatial prediction of eider density in 2013 zoomed in to the Triangle area with no detection correction. Because there is no space-time interaction, relative densities are the same across years. Similarly, the detection correction does not vary across years or other covariates, so the relative surface does not change with or without a detection correction.

of the period (e.g., 6.5, 13.0, 19.5, and so on) so that similar phases of the cycle are compared. For example, a 19-year trend calculated from 2003 to 2022 (approximately trough to trough) gives a posterior mean of 0.03 (95% CI: −0.01, 0.09).

## Discussion

Steller's eider numbers were estimated using generalized additive models and traditional design-based methods for two survey areas, and then using models after combining the two data sets. The Triangle data survey is a small subset of the ACP that contains most of the eiders and is sampled intensively; whereas, the ACP is a much larger area that covers additional areas where Steller's eider have been observed, such as the Teshekpuk area, but is sampled at a lower intensity. Because of the rarity of Steller's eider in all areas but the most northern area of the Triangle, design-based estimates are imprecise and the sampling process can cause increased variance and zero observations in some years. This is especially the case for the ACP survey where few or no Steller's eider exist in most sample strata. Model-based estimates can ameliorate some of these issues by using an explicit probability model for the sampling process and sharing information

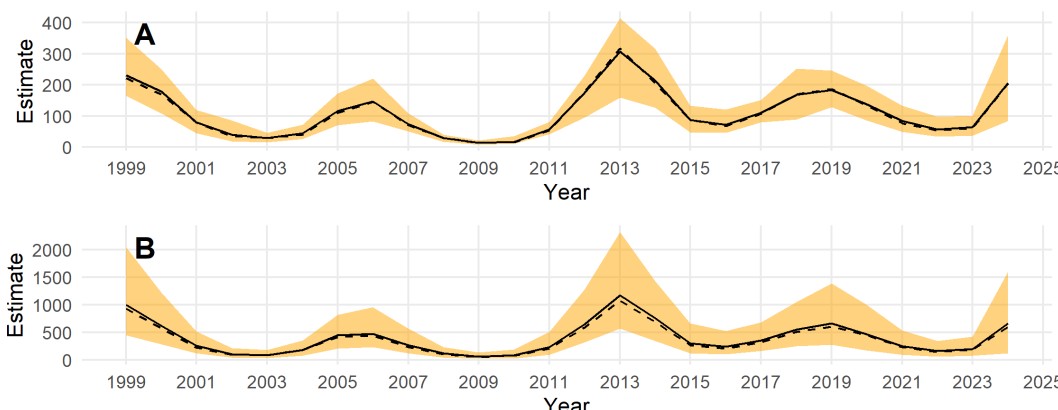

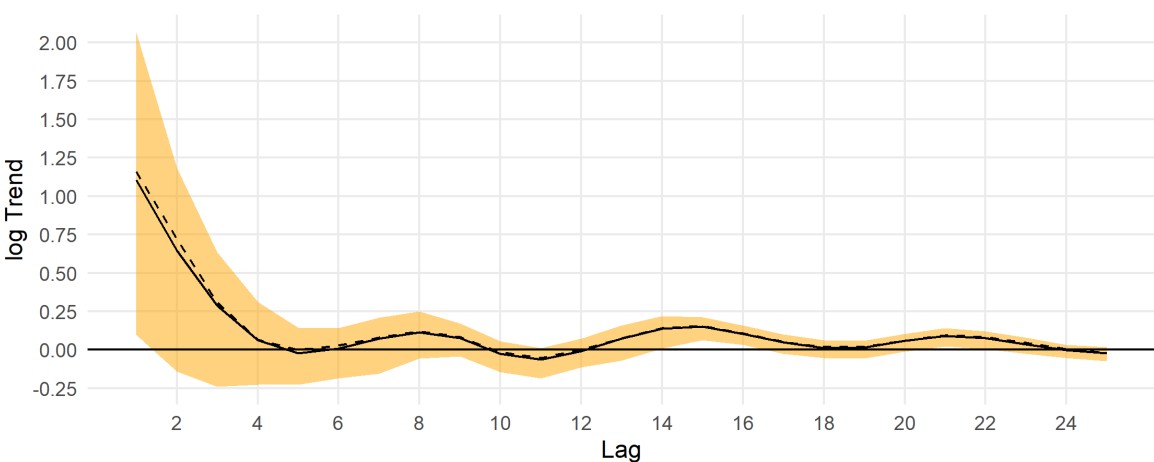

**Fig 13. Aunnual estimates of Steller's eider across the whole Arctic Coastal Plain, 1999-2024.** Estimates are from the best model fit to the Arctic Coastal Plain and Triangle data combined. Posteriors are given **(A)** without accounting for detection and **(B)** after applying a detection correction. The black line is the posterior mean, the dashed black line is the posterior median, and the orange band is the 95% credible interval.

**Fig 14. Posterior trend estimates for Steller's eider on the Arctic Coastal Plain, 1999-2023.** Estimates are from the best model fit using data from both the Triangle and Arctic Coastal Plain surveys. The y axis is the log of the geometric mean growth rate, and the x axis is the lag-year trend, the lag of 10 gives the 10-year trend from 2014 to 2024. The black line is the posterior mean, the dashed black line is the posterior median, and the orange band is the 95% credible interval. The horizontal thick black line is the y-axis origin.

across years and space to smooth estimates. Such smoothing helps to remove sampling variation and detect the spatio-temporal population dynamics, such as the periodic cycles (Fig 13), average spatial variation in density across time (Fig 6), or dynamics in spatial density across time (Fig 5C).

Which results should one use? This depends on the purpose. If one is simply interested in comparing survey results across years, then the design-based estimate can help make rough assessments of changes in the underlying population by comparing estimates and confidence intervals across years. These calculations can also be useful for assessing design properties of the surveys, such as sampling effort and power and are very simple to do. While detection corrections can be applied to design-based estimates, interpreting years of zero observations is problematic, as no detection correction can be made in these years without specifying a probability model that explicitly allows for zero observation when animal are present. If one is interested in population size, population trends, or variation in density across space, then some

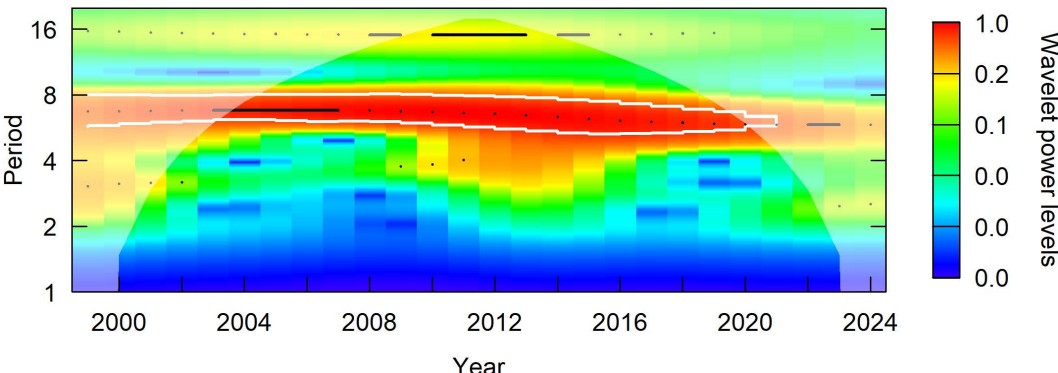

**Fig 15. Wavelet power spectrum for Steller's eider across the Arctic Coastal Plain.** The power spectum was calculated at the posterior mean of the total population estimate. Black points and lines give ridges where wavelet power is a local maximum. White lines give a 95% confidence interval for the period. Larger power levels (red) correspond to greater probability density of the wave period. The 'cone of influence' is shown by the slightly blurred area where boundary effects on estimating wave period are large.

form of probability model that includes "regularization" of estimates should be used. While other models can be used, the GAMs presented here are easily applied, seem to describe the data well, and have all of the advantages discussed above. They are also commonly applied for trend estimation (e.g., [6,10,11]). Another advantage of the linear models used here (GAMs) is that interpretation and understanding is straightforward, where effects can be partitioned out into time, space, and space-by-time interaction effects (e.g., Fig 5).

For inference in the Triangle area, results from the models fit to the Triangle-only data are best. This is for several reasons. First, sampling intensity is high and consistent across space. This affects the amount of smoothing that the GAM will select. Under sparse sampling, a smoother surface will result, all else being equal (i.e., the true wiggliness of the density surface). Compared to results in the Triangle area from the combined model, the smooth of the density surface will reflect the average properties across the whole ACP and this will result in over-smoothing the Triangle area. This can be seen by comparing Fig 6 to Fig 12C. In addition, the high sampling intensity across a relatively long time period has allowed the estimation of a simple space-time interaction effect in the Triangle area where density is increasing in the north and decreasing in the south through time (Fig 5C). The model also estimates the temporal effect well with much better precision than the design-based estimates (compare Fig 5B to Fig 4A).

For inference outside the Triangle or the ACP as a whole, the model using the combined data sets is best because design-based estimates using only the ACP data are extremely imprecise for the Steller's eider (Fig 4B) and even model-based estimates, while much better, are not precise (Fig 10) compared to results from the combined data. Thus, combining data sets allows the Triangle data to inform the temporal estimate, while the ACP data allows estimation of density outside the Triangle area. Comparing Fig 7 to Fig 13 shows that nearly half the population might be found outside the Triangle area in some years. Because the space-time interaction of the Triangle-only model shows a decreasing trend in the south (Fig 5C), there may be fewer eiders outside the Triangle in recent years than there were in earlier years. In any case, including the ACP survey data allows estimation of eider density in the Teshekpuk and other areas of the ACP. If the ACP data is not available, however, then survey effort in the Triangle area only is likely sufficient to monitor abundance and trends of the population, given the current distribution.

The model and approach used here is similar to that of Amundson et al. [6], with two important simplifications. First, the time series for the ACP data was restricted to 2007–2024. This allows a much simpler model because adjusting for survey timing differences pre-2007 is not necessary. Thus, the phenology covariates used in [6] are not included or necessary in the models used here. Second, restricted maximum likelihood as implemented in `mgcv` [39] was used instead of direct

Bayesian simulations (Markov chain Monte Carlo). This is much easier from a development and computational time perspective, making the current approach easy to apply without custom coding in Bayesian software. Instead, these models can be applied in R using more familiar linear model syntax, the model fitting algorithm is much faster than direct Bayesian simulations, and results are easier to assess. Another difference is that the current effort estimated observer effects, which were not in [6], and incorporated a detection estimate. While the observer effects contributed nothing for the Steller's eider (Table 2), they have for other species of waterbirds in separate analyses [2] or other systems [40,41]. Therefore, observer effects are probably also important for the Steller's eider but simply cannot be justified based on parsimony inherent in AIC model selection criteria. The effect of "Survey" in the combined analysis is curious. This suggests a strong effect of observer if there are not large differences in protocol, which there are not, or in implementation between the surveys. This effect deserves more investigation. In any case, for population estimates, predictions were made setting the survey parameter to the Triangle (the intercept) because this is the survey area for which detection was estimated. Because there are so few observations of Steller's eider on the ACP survey, however, this effect may not be estimated well and deserves more investigation. One solution might be to sample the Triangle area with higher intensity during the ACP survey.

One interesting comparison to Amundson et al. [6] is that they found very linear time trends for Steller's eider. This was also the case here when models included a spatio-temporal interaction and were fit to ACP-only data (model M2, Table 2, results at https://doi.org/10.7944/22x5-t856). This model, however, was not well-supported compared to simpler models (Table 2). When fit to data that combined ACP and Triangle data, the model with a spatio-temporal interaction was well-supported but was not used here because it gave widely unbelievable predictions from 1999 to 2006. I attributed this to the spatio-temporal mismatch in data sets during this period and a spatio-temporal mismatch in sampling intensity. This was not a problem in the data of [6], but they did not fit simpler models without a space-time interaction. Because the simpler model without a space-time interaction reported here produced reasonable predictions and all other data sets produced similar cyclic temporal patterns, I believe the simpler model with temporal and spatial effects is a more appropriate model when data sets are combined, and the results of [6] are simply due to a lack of power to detect anything other than linear smooths of year in their more complex model. In any case, a perfectly linear (on the log scale) temporal trend is hard to believe and most likely represents a modelling artifact of this more complex model when confronted with sparse or mismatched data in time and space.

## Conclusion

Steller's eider populations on the ACP, including the Triangle area, are clearly undergoing cyclical patterns of abundance with a period of about 6.5 years (Fig 7, Fig 13, Fig 15). While it has been hypothesized that this is due to lemming cycles [3], this effort cannot determine the cause of the cycles. Lemmings tend to cycle with a shorter period, closer to 3–6 years with a median period around the arctic of 3.7 years [42]. The cycle near the Triangle survey is somewhat unclear from historical data [43], so unless the lemming period is ≥5.5 years, the estimate of periodicity presented here does not support a simple concordance between eider and lemmings or their predators. It is clear that eider cycles exist and that the population has been fluctuating around stable cycles for at least 25 years (Fig 15). Thus, any measure of trend will show an increasing, decreasing, or stable estimate depending on the beginning and ending points for the trend calculation (Fig 8, Fig 14). Recovery criteria and species status assessments should incorporate this understanding into trend determinations. Attention should be given to the start and end points for trends so that similar points in the cycle are used (integer multiples of approximately 6.5). While each cycle is a stochastic realization that will not follow exactly the same period, care should be taken when interpreting estimates of trend. For example, a 10-year trend from 2013 to 2023 would show a large and significant decrease simply due to the cycle phase; whereas, 2009–2019 would show a large increase. Instead, the population seems to be undergoing fairly stable cycles of constant period.

The detection estimate used here (0.307, SD = 0.092) is substantially lower and more variable than used in the Species Status Assessment of 2019 (0.43, SD = 0.028, [44]), which was based on estimates for long-tailed duck (*Clangula*

*hyemalis*) and did not account for non-detection by both observers (a '00' capture history). Analyses previous to 2019 (USFWS, unpublished) used a point estimate of 0.3 (SD = 0) based on expert judgement, which is remarkably close to the mean estimate used here, but did not include any uncertainty. Dunham and Grand [4] used the same point estimate and allowed for some uncertainty (SD = 0.02). Therefore, the posterior mean population estimates reported here will be higher and skewed toward larger values than past estimates. Better estimates of detection could have large effects on the population estimates. Until such estimates are available and population estimates are updated, the current estimates should be treated as the best available, and the approach used here seems a reasonable way to incorporate prior (or posterior) distributions of detection into population estimates.

## Supporting information

**S1 Table. Detection estimates by distance from transect.** This is an unpublished USFWS report that describes the field experiment and analysis to estimate detection probability of Steller's eider by an aerial observer in 2017 and 2018. Table 3 is used to calculate the detection probability averaged over distance.
(DOCX)

## Acknowledgments

Comments from Annie Maliguine, John Nash, Dave Safine, Julian Fischer, Amy Pocewicz, and Chuck Frost, and two reviewers improved the manuscript. Cat Bradley designed and analyzed the initial years of the detection experiment. Tim Obritschkewitsch (ABR, Inc.) provided data, including improvements in data over that available in 2019, and helped to interpret data so that it could be used in this analysis. Many USFWS pilots and observers have collected ACP data over the years, making this analysis possible. The findings and conclusions in this article are those of the author and do not necessarily represent the views of the U.S. Fish and Wildlife Service.

## Author contributions

**Conceptualization:** Erik E. Osnas.

**Data curation:** Erik E. Osnas.

**Formal analysis:** Erik E. Osnas.

**Methodology:** Erik E. Osnas.

**Writing – original draft:** Erik E. Osnas.

**Writing – review & editing:** Erik E. Osnas.

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
