## [Decision Letter · Decision Letter 0]

2 Mar 2026

PONE-D-26-04075Periodicity in Steller's eider (*Polysticta stelleri*) population size and density on the Arctic Coastal Plain, Alaska, revealed using generalized additive models) population size and density on the Arctic Coastal Plain, Alaska, revealed using generalized additive modelsPLOS One

Dear Dr. Osnas,

Thank you for submitting your manuscript to PLOS ONE. I have now heard back from two reviewers who were quite positive about the manuscript and recommend that I move it forward for acceptance for publication following some recommendations for minor revisions, particularly those made by Reviewer 2. Could you please consider the recommendations made by the reviewers and provide a new version of the manuscript that addresses those concerns? 

A letter that responds to each point raised by reviewer(s). You should upload this letter as a separate file labeled 'Response to Reviewers'.A marked-up copy of your manuscript that highlights changes made to the original version. You should upload this as a separate file labeled 'Revised Manuscript with Track Changes'.An unmarked version of your revised paper without tracked changes. You should upload this as a separate file labeled 'Manuscript'.

If applicable, we recommend that you deposit your laboratory protocols in protocols.io to enhance the reproducibility of your results. Protocols.io assigns your protocol its own identifier (DOI) so that it can be cited independently in the future. For instructions see: https://journals.plos.org/plosone/s/submission-guidelines#loc-laboratory-protocols. Additionally, PLOS ONE offers an option for publishing peer-reviewed Lab Protocol articles, which describe protocols hosted on protocols.io. Read more information on sharing protocols at . Additionally, PLOS ONE offers an option for publishing peer-reviewed Lab Protocol articles, which describe protocols hosted on protocols.io. Read more information on sharing protocols at https://plos.org/protocols?utm_medium=editorial-email&utm_source=authorletters&utm_campaign=protocols..

We look forward to receiving your revised manuscript. Thanks again for choosing to communicate your scientific findings in PLOS ONE.

Kind regards,

Lee W Cooper, Ph.D.

Section Editor

PLOS One

Journal Requirements:

3. We noted in your submission details that a portion of your manuscript may have been presented or published elsewhere. “No, but this analysis has been included an unpublished government report available online at https://doi.org/10.7944/3vzp-0r93”

4. We note that Figure 1 and 2 in your submission contain map images which may be copyrighted. All PLOS content is published under the Creative Commons Attribution License (CC BY 4.0), which means that the manuscript, images, and Supporting Information files will be freely available online, and any third party is permitted to access, download, copy, distribute, and use these materials in any way, even commercially, with proper attribution. For these reasons, we cannot publish previously copyrighted maps or satellite images created using proprietary data, such as Google software (Google Maps, Street View, and Earth). For more information, see our copyright guidelines: http://journals.plos.org/plosone/s/licenses-and-copyright.

1. You may seek permission from the original copyright holder of Figure 1 and 2  to publish the content specifically under the CC BY 4.0 license.

5. Please include a copy of Table 3 which you refer to in your text on page 12.

Reviewers' comments:

Reviewer's Responses to Questions

**Comments to the Author**

1. Is the manuscript technically sound, and do the data support the conclusions?

Reviewer #1: Yes

Reviewer #2: Yes

2. Has the statistical analysis been performed appropriately and rigorously?

Reviewer #1: Yes

Reviewer #2: Yes

3. Have the authors made all data underlying the findings in their manuscript fully available?

The PLOS Data policy requires authors to make all data underlying the findings described in their manuscript fully available without restriction, with rare exception (please refer to the Data Availability Statement in the manuscript PDF file). The data should be provided as part of the manuscript or its supporting information, or deposited to a public repository. For example, in addition to summary statistics, the data points behind means, medians and variance measures should be available. If there are restrictions on publicly sharing data—e.g. participant privacy or use of data from a third party—those must be specified. requires authors to make all data underlying the findings described in their manuscript fully available without restriction, with rare exception (please refer to the Data Availability Statement in the manuscript PDF file). The data should be provided as part of the manuscript or its supporting information, or deposited to a public repository. For example, in addition to summary statistics, the data points behind means, medians and variance measures should be available. If there are restrictions on publicly sharing data—e.g. participant privacy or use of data from a third party—those must be specified.

Reviewer #1: Yes

Reviewer #2: Yes

4. Is the manuscript presented in an intelligible fashion and written in standard English?

Reviewer #1: Yes

Reviewer #2: Yes

5. Review Comments to the Author

Reviewer #1: I am not experienced in the use of spatio-temporal generalized additive models, whose presentation is perhaps the most important contribution of this paper to the general readership of PLoS ONE. I suggest that alternative reviewers with greater expertise in this area should also be consulted. However, the methods are described quite thoroughly and intuitively, and the people listed in the Acknowledgements who have already reviewed the paper are well qualified in terms of relevant statistical experience and familiarity with the study system.

My assessment is that this is a very carefully prepared paper, in which sophisticated statistical analyses are presented in a readable and understandable way. The figures are effective and comprehensive. Thorough comparison of the new and traditional methods is quite valuable, given that the traditional methods have been used in multiple venues and will probably be continued to maintain time series. The novel analyses of cyclic patterns, verified by two different analytical approaches, are important to future evaluation of trends for both the target species and the many others analyzed by the design-based estimates. These analyses also provide new insights into potential lemming cycles in this study area and future efforts to characterize and understand them. The spatial trends in eider distributions will be helpful in management planning.

The paper is well-enough written that, uncharacteristically, I have few suggestions to make. Essentially all my comments are trivial grammatical corrections.

1. L 117. Please replace “the above” by “equation (2)”

2. L 351. Replace “miss-identification” by “misidentification”

3. L 360. Delete “and” before “Medium”

4. L 462. Replace “spectorgram” by “spectrogram”

5. L 465. Replace “ocilation” bny “oscillation”

6. L 577. Replace “miss-matched” by “mismatched”

Reviewer #2: This study demonstrates that generalized additive models provide more precise and consistent estimates of Steller’s eider population size and density in Alaska than traditional design-based methods. The results also highlight significant cyclical fluctuations in population size which have strong implications for interpreting trend estimates used for population recovery. Additionally, density analyses provide further evidence of high concentrations in the northern part of the survey area and indications of northward shifts in density. These findings offer improved methods for assessing population trends and recovery of Steller’s eiders, a problem that has made decision-making challenging in the past. The author appropriately cited the relevant literature both on the species and modeling methods and provided sufficient context for non-specialist readers. The data and analyses are robust, appropriately support the claims, and are well-organized and easy to access via the linked github repository. Methods are sufficiently detailed to reproduce the analysis (especially including the detailed code). The manuscript is well-written but see minor comments below for typos and other small suggestions. I have no major issues or comments for the author to address. Note: I did not run all of the code but the files that I opened looked organized and were easy to access.

Specific Comments:

• Lines 113-119: The concerns highlighted here may warrant further explanation – it’s not made clear here exactly why the estimator is better than the ratio estimator if there are issues in estimation with a strong density gradient. Given the system and the assumptions of a ratio estimator I agree that this is likely better, but this section could use work to explain why including citations if possible. Additionally, the last sentence includes “case” twice, I would suggest the second one be changed to “specific system”.

• Line 465: typo, “ocilation” . “oscillation”

• Line 564: “When” seems to start with a bolded “W” – this occurs in a couple of other locations but it may be a rendering problem in adobe.

• Figures

o Figures 1 and 2 show the same area with the same distance but are different sizes – consider constraining the output size to be consistent.

o Figure 3 may be easier to read as a “lollipop” chart and better y axis – but not necessary.

o Figure 4 is the only two-part figure that shows up together and has A/B labels within the plot. Check all of the two-part and three-part figures for consistency before resubmission.

o Figure 9B doesn’t seem necessary for communicating any results – consider removing or emphasizing the purpose of including it.

o Figure 15 “period” is really far from the numbers of the y-axis. Might be easier to see what is happening if the x-axis had more years but turn then 45 or 90 degrees.

o The trends/abundance figures may benefit from reducing the background noise, specifically, reducing the number of gridlines, making the background white, etc. Just a style choice but it would be easier to read.

6. PLOS authors have the option to publish the peer review history of their article (what does this mean?). If published, this will include your full peer review and any attached files.). If published, this will include your full peer review and any attached files.

**Do you want your identity to be public for this peer review?** For information about this choice, including consent withdrawal, please see our  For information about this choice, including consent withdrawal, please see our Privacy Policy..

Reviewer #1: No

Reviewer #2: No

---

## [Author Response · Author response to Decision Letter 1]

26 Mar 2026

Response to Reviewers

Dear Editor,

Thank you and thanks to the reviewers for providing the reviews and formatting guidance. I have revised the manuscript as suggested. I have also revised all figures by (1) making all multi-panel figures one image and (2) re-formatting each figure as a tiff file. This significantly changed Fig 15, which now shows more regions of the spectrogram. Just to be clear, I have not changed any analysis for Fig 15, only the file format. In the original image in eps format, the shaded region could not be displayed, I believe because this region is a separate “raster layer” and eps format can only show one layer. The new figure is much better. In terms of other review comments, I have accepted all of them and made the needed changes.

In addition to the typos and revisions suggested by the reviewers, I have also found a few more typos and made a few other small revisions, as detailed in the track changes version of the PDF.

Below I respond to your comment under “Journal Requirements” and then respond to each comment of the reviewers, with my response in italics.

Thank you for the opportunity to revise this manuscript and for your efforts.

Erik

Comments under “Journal Requirements”:

In Comment #1 about style requirements, I have thoroughly revised the manuscript and accompanying .tex file to adhere to your style requirements. The original manuscript and .tex file was produced using the PLOS Latex extension for Quarto, which is based on your latex template. During this revision, I tried to use your template directly but could not get it to compile. I am not a Latex expert, so I can’t provide any reasons for this, it is probably due to my lack of knowledge. By visual inspection, the revised PDF appears to follow your guidelines. Note that there are some small discrepancies between your provided Latex template and the online “Affiliations guidance”. Please let me know if there is more to do.

In Comment #2 on code sharing, I have made all code used for this report available at https://github.com/USFWS/STEI-estimates ,as noted by the reviewers. Upon acceptance of the manuscript, I can do a “code release” on GitHub where the code version for this manuscript is packaged together as a downloadable zip file and associated with the publication.

In Commet#3 you asked for clarification about peer-review and “formal publication” of a pervious report that the current manuscript is based on. The previous report can be found at https://doi.org/10.7944/3vzp-0r93. This previous report has never undergone external peer-review, as has the current manuscript, and was only posted to our (USFWS) data repositories catalog (we use the USGS ScienceBase). It is not clear to me what is meant by “formal publication”, but I would consider posting the report to our web-catalog equivalent to posting to a pre-print server, as PLOS encourages authors to do. In addition, the current manuscript has been significantly shortened and revised over the posted report.

In Comment #4 about copyrighted map material (Fig 1 and 2), you noted that I must obtain permission to publish these or use public domain data for maps. Please note that all map data used for my figures are US Government works and are in the Public Domain. Specifically, the survey area polygons were created by us (the US Fish and Wildlife Service) and the outline of Alaska was sourced from the open-source R package spData, which sources the data from the US Census Bureau. I have added the following to the captions of figure 1 and 2:

All map data are products of the U.S. Government and are in the Public Domain.

In Comment #5 on Table 3, please note that “Table 3” refers to the Table 3 in the Supplemental material. I revised the text in the manuscript to read “Table 3 in S4 Detection”.

Your Comment #6 is not applicable. The reviewers did not suggest any additional citations.

In Comment #7 on the reference list. There have been no changes during revision. I have not cited any retracted papers. I believe it to be complete.

Response to reviewers (my response is in italics)

Reviewer 1:

1. L 117. Please replace “the above” by “equation (2)”

Revised.

2. L 351. Replace “miss-identification” by “misidentification”

Revised.

3. L 360. Delete “and” before “Medium”

Revised.

4. L 462. Replace “spectorgram” by “spectrogram”

Revised.

5. L 465. Replace “ocilation” bny “oscillation”

Revised.

6. L 577. Replace “miss-matched” by “mismatched”

Revised.

Reviewer 2:

Specific Comments:

• Lines 113-119: The concerns highlighted here may warrant further explanation – it’s not made clear here exactly why the estimator is better than the ratio estimator if there are issues in estimation with a strong density gradient. Given the system and the assumptions of a ratio estimator I agree that this is likely better, but this section could use work to explain why including citations if possible. Additionally, the last sentence includes “case” twice, I would suggest the second one be changed to “specific system”.

I agree and have revised the paragraph to include the specific reason the ratio estimator fails in this case. I had originally neglected to say that the advantage of the ratio estimator depends on a positive correlation between the two variables (here animal count and transect length/area). In this survey region, most animals are observed in the north on shorter transects (a consequence of the animal distribution, triangle shape of the area, and east-west transects). Therefore, there is a negative correlation between transect length and animal count and the ratio estimator does very poorly (i.e., has high variance) compared to other estimators. Therefore, I use a line transect estimator commonly used in distance sampling (Fewster). Here is the revised first half of the paragraph leading up to Equation (1) and (2):

I calculated a design-based estimate using formula R3 of Fewster et al. [24], (also see [25], p.79), modified for strip transects. A ratio estimator, which is more common for strip-transect surveys [26, 27, 28], was investigated but found to be poor (very high variance) because most observations of Steller's eiders are on short transects in the northern end of the triangular-shape area and longer transects in the south rarely have observed eiders. The ratio estimator can provide lower variance estimates when there is a positive correlation between the two variables (here, animal count and transect length or area) [26, 28]. In this survey area, however, there is a negative correlation between count and transect length (or area). This makes the ratio estimator worse than a simple plot-based estimates or the "R3" estimate of Fewster we use here [24].

• Line 465: typo, “ocilation” . “oscillation”

Revised.

• Line 564: “When” seems to start with a bolded “W” – this occurs in a couple of other locations but it may be a rendering problem in adobe.

Interesting. I see the same effect in the PLOS-formatted version without figures but the effect is not in the supplemental version I submitted that included figures in the text. I have checked the .tex document, and no special formatting is applied to this character. I assume this is an Adobe issue and can be dealt with during copy editing if necessary. During the revision, this same effect appears in the compiled PDF when viewed in Adobe but not in the RStudio PDF viewer, so I assume this is in fact an Adobe issue as the reviewer suggested.

• Figures

o Figures 1 and 2 show the same area with the same distance but are different sizes – consider constraining the output size to be consistent.

I have revised this and all other figures. These two figures are now the same size.

o Figure 3 may be easier to read as a “lollipop” chart and better y axis – but not necessary.

I agree. The figure is now a “lollipop” chart with a slight offset between the two surveys. Thanks for the suggestion.

o Figure 4 is the only two-part figure that shows up together and has A/B labels within the plot. Check all of the two-part and three-part figures for consistency before resubmission.

I have revised this and all figures so that each is one image file with labels created in the figure image.

o Figure 9B doesn’t seem necessary for communicating any results – consider removing or emphasizing the purpose of including it.

I agree. Panel B of Figure 9 has been removed. The time series is now displayed only in Figure 10, which is sufficient to show the random effects on the response scale.

o Figure 15 “period” is really far from the numbers of the y-axis. Might be easier to see what is happening if the x-axis had more years but turn then 45 or 90 degrees.

I increased the number of year tick marks and labels and re-sized the figure. I was able to add year labels without needing to turn them from the horizontal. I moved the “Period” title a little closer to the y axis. Note that in the new figure in tiff format, areas of the spectrogram outside the “cone of influence” (slightly shaded) are now shown. In the original submitted figure (eps format) this area was dropped, I believe because eps format cannot handle more than one raster layer in an image. For submission, I did not understand this difference between eps and tiff formats, and I believe the revised (tiff) figure is better than the eps version.

o The trends/abundance figures may benefit from reducing the background noise, specifically, reducing the number of gridlines, making the background white, etc. Just a style choice but it would be easier to read.

I removed the grey background (ggplot theme_minimal) and the minor grid lines in all figures.

---

## [Editor Report · Decision Letter 1]

29 Mar 2026

Periodicity in Steller's eider (*Polysticta stelleri*) population size and density on the Arctic Coastal Plain, Alaska, revealed using generalized additive models) population size and density on the Arctic Coastal Plain, Alaska, revealed using generalized additive models

PONE-D-26-04075R1

Dear Dr. Osnas,

Thank you for submitting a revised version of your manuscript and addressing the minor issues identified by the reviewers. I also have to credit the editorial office for asking you to address the copyright, formatting, and other technical issues associated with the manuscript. I'm pleased to let you know that your manuscript has been judged scientifically suitable for publication and will be formally accepted for publication once it meets any remaining technical requirements that the editorial office identifies.

Within one week, you’ll receive an e-mail detailing those required amendments. When these have been addressed, you’ll receive a formal acceptance letter and your manuscript will be scheduled for publication.

An invoice will be generated when your article is formally accepted. Please note, if your institution has a publishing partnership with PLOS and your article meets the relevant criteria, all or part of your publication costs will be covered. Please make sure your user information is up-to-date by logging into Editorial Manager at Editorial Manager® and clicking the ‘Update My Information' link at the top of the page. For questions related to billing, please contact  and clicking the ‘Update My Information' link at the top of the page. For questions related to billing, please contact billing support..

We would welcome additional publicity associated with publication of your findings, so please advise the appropriate people within the USFWS about your upcoming paper to help maximize its impact. If they’ll be preparing press materials, please inform our press team as soon as possible -- no later than 48 hours after receiving the formal acceptance. Your manuscript will remain under strict press embargo until 2 pm Eastern Time on the date of publication. For more information, please contact onepress@plos.org.

Thank you again for choosing to publish through PLOS ONE and congratulations on the pending acceptance of your paper.

Kind regards,

Lee W Cooper, Ph.D.

Section Editor

PLOS One

---

## [Editor Report · Acceptance letter]

PONE-D-26-04075R1

PLOS One

Dear Dr. Osnas,

I'm pleased to inform you that your manuscript has been deemed suitable for publication in PLOS One. Congratulations! Your manuscript is now being handed over to our production team.

Kind regards,

on behalf of

Dr. Lee W Cooper

Section Editor

PLOS One